



# Reconstructing Younger Dryas Ground Temperature and Snow Thickness from Cave Deposits

Paul Töchterle[1], Anna Baldo[1], Julian B. Murton[2], Frederik Schenk[3,4], R. Lawrence Edwards[5], Gabriella Koltai[1] and Gina E. Moseley[1]

[1]Institute of Geology, University of Innsbruck, Innsbruck, Austria
[2]Department of Geography, University of Sussex, Brighton, UK
[3]Department of Geological Sciences and Bolin Centre for Climate Research, Stockholm University, Stockholm, Sweden
[4]Department of Geosciences and Geography, University of Helsinki, Helsinki, Finland
[5]Department of Earth Sciences, University of Minnesota, Minneapolis, USA

*Correspondence to*: Paul Töchterle (paul.toechterle@uibk.ac.at)

**Abstract.** The Younger Dryas Stadial was characterized by a rapid shift towards cold-climate conditions in the North Atlantic realm during the last deglaciation. While some climate parameters including atmospheric temperature and glacier extent are widely studied, empirical constraints on permafrost temperature and snow thickness are limited. To address this, we present a regional dataset of cryogenic cave carbonates (CCCs) from three caves in Great Britain that formed at temperatures between -2 and 0 °C. Our CCC record indicates that these permafrost temperatures persisted for most of the Younger Dryas. By combining ground temperatures with surface temperatures from high-resolution, ground-truthed model simulations, we demonstrate that ground temperatures were approximately 6.6 ±2.3 °C warmer than the mean annual air temperature. Our results suggest that the observed temperature offset between permafrost and the atmosphere can be explained by an average snow thickness between 0.2 and 0.9 m, which persisted for 233 ±54 days per year.

## Introduction

The Younger Dryas Stadial refers to a period from approximately 12.9 to 11.6 ka BP (Cheng et al., 2020) when the climatic transition from the Last Glacial Maximum to the Holocene was interrupted by a shift towards extremely cold-climate conditions in the North Atlantic realm. Proxy records from northwest Europe and Greenland show the most severe cooling (e.g., Renssen and Isarin, 1997; Heiri et al., 2014), whereas temperature trends in parts of the southern hemisphere and elsewhere were attenuated or even reversed (e.g., Shakun and Carlson, 2010; Carlson, 2013; Reeves et al., 2013). There is ongoing debate on the cause of the Younger Dryas. The most prominent hypotheses include a slow-down of the Atlantic Meridional Overturning Circulation (AMOC) in response to increased meltwater input (e.g., Broecker, 2006; Murton et al., 2010), atmospheric reorganisation (e.g., Brauer et al., 2008), radiative forcing after an extra-terrestrial impact (e.g., Firestone et al., 2007; Israde-Alcántara et al., 2012) and a combination of these factors (Renssen et al., 2015).



In the North Atlantic realm, the Younger Dryas was characterized by extreme seasonal changes: relatively short summers that likely were only slightly cooler than those of today contrasted with long and cold winters (Denton et al., 2005; Schenk et al., 2018). Sea level was approximately 60 m lower than at present (e.g., Smith et al., 2011; Cronin, 2012), exposing large parts of the European continental shelf that today are covered by the English Channel and the North Sea (**fig. 1**). In the local chrono-stratigraphic nomenclature of the British Isles, the Younger Dryas is traditionally seen as the equivalent to Loch Lomond

Stadial, which has primarily been defined by moraine chronologies (e.g., Ballantyne and Harris, 1994; Golledge et al., 2007; Golledge, 2010). Recent studies have cast doubt on the synchronicity of the Loch Lomond Stadial with the Younger Dryas, arguing that glaciers already formed during the preceding interstadial and melted during the Younger Dryas (Bromley et al., 2014; Putnam et al., 2019; Bromley et al., 2018; Bromley et al., 2023). Nevertheless, a majority of studies on independent proxy records dated to the Younger Dryas infer a significantly colder climate and lower mean annual air temperature (MAAT)

than today (e.g., Swabey, 1996; Jones et al., 2004; Palmer et al., 2010; Brooks and Langdon, 2014; Lincoln et al., 2020; Timms et al., 2021).

While many components of the Younger Dryas climate system of the British Isles have been inferred from proxy data, few studies have addressed past permafrost (Ballantyne and Harris, 1994; Isarin, 1997; Williams, 1965; Murton and Ballantyne, 2017), in part due to limitations of identifying and dating evidence of the former permafrost presence. Reliable archives that

provide direct evidence for past permafrost include ice-wedge pseudomorphs and a variety of geomorphological features (e.g., French, 1999; Vandenberghe et al., 2012; Vandenberghe et al., 2014; Ballantyne, 2018a). Hiatuses in the growth periods of common speleothem deposits like stalagmites and flowstones have also been used to reconstruct past permafrost at mid- and high-latitudes (Vaks et al., 2013; Vaks et al., 2020; Biller-Celander et al., 2021). However, a weakness of this approach is that it merely implies hydrological inactivity of the karst aquifer, which may or may not be related to sub-zero temperatures and

permafrost. Consequently, additional context is required to use hiatuses in common speleothems as an indicator for past permafrost.

Cryogenic cave carbonates (CCCs) are a special subtype of speleothem that occur as loose piles of morphologically diverse carbonate crystals in isolated cave chambers and are commonly associated with permafrost (Žák et al., 2004; Žák et al., 2008; Žák et al., 2018). The formation of these mineral deposits with grain sizes larger than approx. 1 mm requires a sufficiently

large volume of water to freeze at a slow rate. Consequently, the cave environment must maintain the solidus temperature of water (i.e., 0 °C or slightly below that if the solution is sufficiently briny) for extended periods of time. Previous studies have defined the CCC formation window between -1 and 0 °C, whereas a temperature between -0.5 and 0 °C is deemed more likely (Koltai et al., 2021; Spötl et al., 2021). Recent studies reported in-situ finds of CCCs and provided further support for this genetic model of slowly freezing ionized water at temperatures slightly below 0 °C , which previously had relied on indirect

field evidence and the carbon and oxygen isotopic composition (Bartolomé et al., 2015; Munroe et al., 2021). CCC formation

conditions are thus well constrained, thereby enabling their use as a reliable paleo-thermometer from which past permafrost can be determined (e.g., Žák et al., 2012; Orvošová et al., 2014; Dublyansky et al., 2018).

In this study, we apply a regional, model-integrated approach to CCCs to elucidate past permafrost distribution and the ground thermal regime. We present a dataset from three caves in the British Isles comprising 10 distinct CCC deposits that date to the

Younger Dryas Stadial. Ground temperatures derived from CCCs are placed in context with concurrent common speleothem deposition in the region, air temperature proxy data and high-resolution climate model simulations. From these data, quantitative constraints on the ground thermal regime as well as the thickness of snow cover are derived. Our results amend previous reconstructions of permafrost and winter precipitation in the British Isles during the Younger Dryas and provide suggestions for the future use of CCCs as a paleo-permafrost archive.

**Study Sites**

The CCC samples presented in this study were taken from three different cave sites across a 300 km-long north–south transect of Great Britain (**fig. 1**). The caves are situated within thick (up to 600 m) sequences of Palaeozoic limestones in the regions of the Peak District east of Manchester (mean elevation ~200 m above sea level (a.s.l.)) and the Mendips Hills south of Bristol (mean elevation ~150 m a.s.l.).

The entrance to Water Icicle Close Cavern (WIC, local topography **fig. S1**, cave plan **fig. S2**) is situated at 341 m a.s.l. No natural entrance is known, and the cave is accessed via a medieval mine shaft. Over 1 km of passages has been explored, most of which formed along a horizontal level between 30 and 40 m below the surface. The present-day MAAT in the region is 8.3 °C and mean annual precipitation amounts to approximately 1000 mm, a negligible portion of which falls as snow (**fig. S3**, ERA5 reanalysis data from 1990 to 2020; Muñoz Sabater, 2019). Monitoring of the cave atmosphere between 2019 and 2020

revealed a stable microclimate with a constant cave air temperature of 8.7 °C (**fig. S4**), in close agreement with the local MAAT. Broken speleothems from this cave have previously been used to identify a period of cave glaciation between 87 and 83 ka BP related to permafrost (Gunn et al., 2020) and two CCC deposits of Younger Dryas age have been reported (Töchterle et al., 2022).

Second, Nettle Pot (NTL, local topography **fig. S1**, cave plan **fig. S2**), located in the Peak District, is a more vertical cave

system with passages as deep as 170 m below the entrance, which lies at 467 m a.s.l. The modern climate is characterized by a MAAT of 8.2 °C and 1040 mm of mean annual precipitation (**fig. S3**). The CCC deposit presented herein is located within a sloping passage at a depth of 130 m. Cave access is technically demanding and precluded long-term monitoring. Due to the depth of the system, however, a thermal anomaly by convective heat fluxes, which could potentially lead to sub-zero temperature at the CCC site in the absence of regional permafrost, is unlikely.





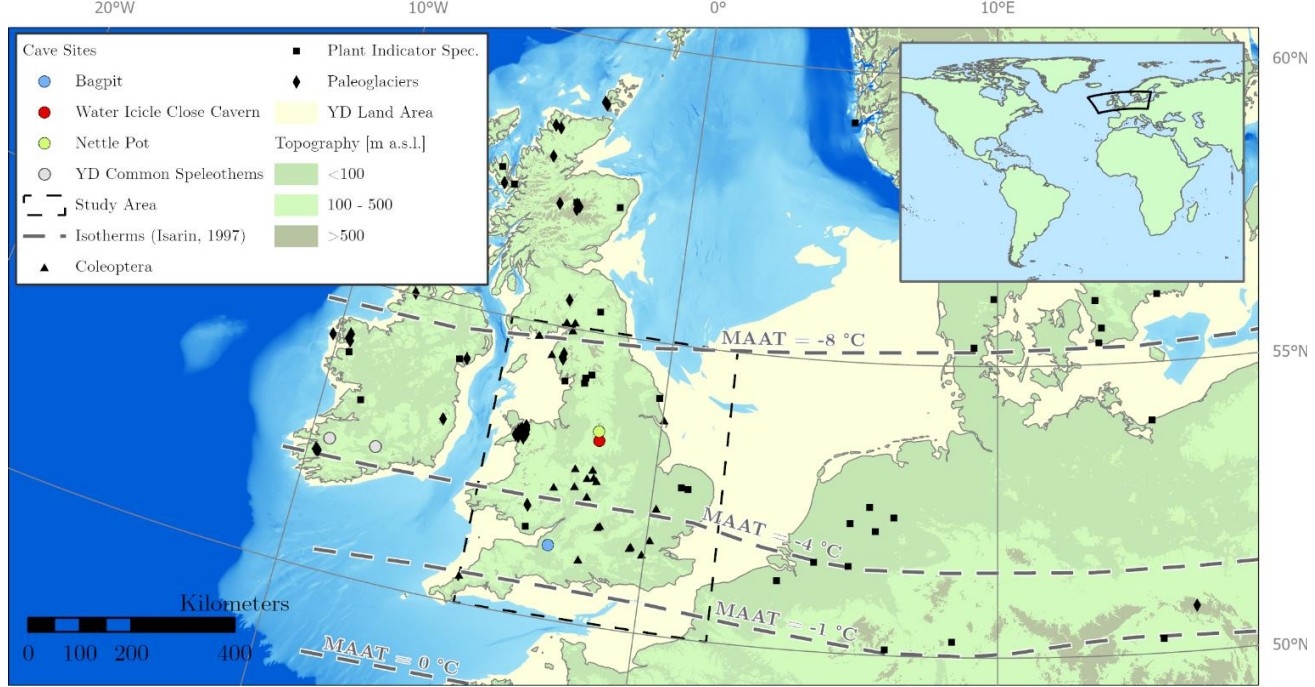

**Fig. 1: Topographical map of the study region. The studied cave sites are marked by coloured dots and two additional caves with reported common speleothem deposition during the Younger Dryas (Swabey, 1996) are shown in grey. The extent of the study area (-5 to 2 °E, 50 to 55 °N) is delineated by a black dashed line. Black symbols depict the study sites that were considered in Younger Dryas surface temperature reconstructions (i.e., triangles - Coleoptera (Atkinson et al., 1987); different biogenic proxies including plant indicator species (Schenk et al., 2018); paleo-glacier ELA (Rea et al., 2020). Dashed isotherms are taken from a temperature reconstruction based on periglacial features (Isarin, 1997). The Younger Dryas land area is equivalent to the -60 m bathymetry contour line and has not been adjusted for post-glacial isostatic rebound.**

Third, Bagpit cave (BGP, local topography **fig. S1**, cave plan **fig. S2**) is a relatively small cave system with a 172 m-long network of explored passages, located in the Mendips Hills south of Bristol. The cave entrance lies at 257 m a.s.l. and was only recently discovered through excavation efforts by caving enthusiasts. The local climate is comparable to the caves in the Peak District, albeit with a warmer MAAT of 10.2 °C and a lower mean annual precipitation of approximately 850 mm. Cave monitoring reveals a seasonal temperature cycle at 15 m below the surface with an amplitude of 0.9 °C and a 6-month phase

shift compared to surface temperatures (**fig. S5**). This indicates that conductive heat flux is the dominant mode of energy transfer in the system and advective heat flux by air ventilation and/or groundwater flow is negligible. These low-amplitude temperature variations were observed at a shallow part of the cave, and they are likely to be attenuated in the deeper parts where CCC samples were taken (between 20 and 28 m below the surface). As with WIC, the mean cave temperature in Bagpit (10.4 °C) closely resembles the local MAAT.

All three caves are complex systems where large parts were previously isolated from the surface and only became accessible through digging and excavation in modern times. The sites where CCCs were found are situated in the homotrophic zone of

the systems, far away from the cave entrance, where tempertures are in stable equilibrium with the local MAAT as evidenced by the monitoring data. These thermal characteristics are a prerequisite for cave-based paleoclimate studies in general and inspire confidence that the selected cave sites did not behave as cold-air traps in the past. The caves all lie on NE-facing slopes,

leeward of the predominant SW wind direction, with relatively gently slope angles below 15° (**fig. S1**). The bedrock is covered with shallow (typically less than 1 m) soil that is drained efficiently through the underlying karst aquifers, leading to a lack of surface water bodies in the immediate vicinity of the caves.

## Materials and Methods

### Monitoring and Sampling

The CCC samples for this study were collected during several field trips between 2018 and 2020. Sites of CCC occurrences were mapped and a subsample was taken from each patch. Sampling of CCC patches was done by scooping up a representative sample with a knife while taking care to leave the overall appearance of the site intact. At two of the three caves (BGP and WIC), temperature loggers (HOBO Water Temp Pro v2, calibrated in an ice bath) were installed and recorded the ambient cave air temperature for at least one year at a temporal resolution of 30 minutes.

### Stable Isotopes

To confirm the cryogenic origin of the samples presented in this study, we used carbon and oxygen stable isotope analysis as a diagnostic tool. From each sample, several specimens were selected and cleaned in an ultrasonic bath with deionised water. After subsequent drying, approx. 20–30 µg of powdered material were drilled from random places on the specimen using a burr-tipped carbide dental drill. Powder samples were then analysed on a Thermo Scientific Delta V Plus mass spectrometer

coupled to a Gas Bench II module according to standard analytical procedures (Spötl and Vennemann, 2003). Results are reported in reference to the VPDB standard at a long-term analytical precision of 0.05‰ for $\delta^{18}O$ and 0.06‰ for $\delta^{13}C$ (Spötl, 2011).

### Dating of CCCs

An aliquot of 3 to 8 CCC specimens (between 20 and 60 mg each) was selected for U/Th analysis at the University of

Minnesota's Trace Metal Laboratory. Where the grain size of CCCs was too small to yield enough material for analysis from a single grain, multiple smaller grains of similar morphology were combined (WIC 2 and WIC 5 in **table 1**). Each aliquot was cleaned in an ultrasonic bath of deionised water prior to analysis, and visible detritus was removed with a toothbrush. The cleaned specimens were dissolved in dilute $HNO_3$ before undergoing chemical preparation (Edwards et al., 1987) and extracts of uranium and thorium were analysed on a Thermo Fisher Neptune Plus multi-collector inductively coupled plasma mass

spectrometer in peak-jumping configuration with an electron multiplier (Shen et al., 2012).





**Table 1: Isochron ages and metrics of the analysed CCC deposits. The reported error on the isotopic ratios corresponds to the respective 2σ uncertainty. The reported R² corresponds to the respective regression in the $^{232}$Th/$^{238}$U vs. $^{230}$Th/$^{238}$U plane of the 3D isochron. Isochron ages are relative to 1950 AD. A graphic representation of the isochrons is provided in the supplementary material.**

| ID | n | ($^{230}$Th/$^{232}$Th)$_{initial}$ [activity] | ($^{230}$Th/$^{238}$U)$_{initial}$ [activity *$10^{-3}$] | δ$^{234}$U$_{initial}$* | R² | Isochron Age (ka BP) |
|---|---|---|---|---|---|---|
| BGP 1 | 7 | 0.89 ± 0.06 | 154.7 ± 0.5 | 377.2 ± 1.1 | 0.9814 | **13.00 ± 0.05** |
| BGP 2 | 7 | 0.4 ± 0.2 | 140.0 ± 0.6 | 262.3 ± 1.2 | 0.4517 | **12.80 ± 0.07** |
| BGP 3 | 3 | 2.4 ± 0.6 | 139.8 ± 3.3 | 253.4 ± 2.2 | 0.9678 | **12.9 ± 0.3** |
| BGP 4 | 4 | 1.41 ± 0.08 | 156.1 ± 0.3 | 468.3 ± 7.0 | 0.9944 | **12.26 ± 0.05** |
| NTL 1 | 5 | 4.2 ± 0.6 | 202.2 ± 0.8 | 998.3 ± 3.6 | 0.9465 | **11.67 ± 0.05** |
| WIC 2 | 3 | 5.4 ± 0.3 | 172.2 ± 1.0 | 575.2 ± 0.3 | 0.9967 | **12.64 ± 0.08** |
| WIC 5 | 3 | 5.03 ± 0.01 | 157.94 ± 0.01 | 564 ± 69 | 0.99999 | **11.6 ± 0.5** |
| WIC 6** | 6 | 7.3 ± 0.7 | 207.7 ± 0.7 | 1076.6 ± 0.8 | 0.9602 | **11.52 ± 0.04** |
| WIC 8** | 8 | 3.4 ± 0.1 | 155.4 ± 1.0 | 507.6 ± 1.2 | 0.9946 | **11.86 ± 0.09** |
| WIC 9 | 5 | 4.1 ± 0.4 | 161.0 ± 2.0 | 509.7 ± 1.5 | 0.9689 | **12.3 ± 0.2** |

* δ$^{234}$U = (($^{234}$U/$^{238}$U) -1) * 1000
** from (Töchterle et al., 2022)

The decay constants used for age calculation were $\lambda_{238}$ = 1.55125 x $10^{-10}$ a$^{-1}$ (Jaffey et al., 1971) and $\lambda_{234}$ = 2.82206 x $10^{-6}$ a$^{-1}$ (Cheng et al., 2013) for uranium and $\lambda_{230}$ = 9.1705 x $10^{-6}$ a$^{-1}$ (Cheng et al., 2013) and $\lambda_{232}$ = 4.9475 x $10^{-11}$ a$^{-1}$ (LeRoux and Glendenin, 1963) for thorium. Reported ages are based on linear $^{234}$U/$^{238}$U-$^{230}$Th/$^{238}$U-$^{232}$Th/$^{238}$U isochron models (Töchterle et al., 2022) and reported in kiloyears before 1950 (ka BP) at a 2σ uncertainty interval of the respective regression.

**Modern Climate Data**

In order to compare our paleo archive with modern analogues, we used the monthly ERA5 Land reanalysis dataset (Muñoz Sabater, 2019) and extracted gridded time series (0.25° x 0.25°) for soil temperature, snow thickness and 2 m air temperature between 1990 and 2020. For permafrost extent and percentages, the 2010 time-slice from ESA Permafrost Climate Change Initiative Extent was used, which is based on satellite data in combination with a thermal model (Obu et al., 2021). We classified the percentage areal extent of permafrost based on generally accepted intervals (i.e., <10 % of area = isolated discontinuous permafrost, 10–50 % = sporadic discontinuous, 50–90 % = extensive discontinuous, >90 % = continuous permafrost). Areas with a permafrost fraction below 10 % were removed from the dataset. For all subsequent calculations, the reanalysis data and the permafrost grid were down-sampled to a lower resolution of 92 km x 92 km using spatial mean values.

**Results and Discussion**

The CCC samples presented in this study were taken from three different cave sites across a 300 km-long north–south transect across Britain (**fig. 1**). The studied samples comprise a large variety of crystal morphologies, as is typical for CCCs. Common morphological types include i) milky-white (semi-) spherical crystals with high degrees of non-crystallographic branching (Shtukenberg et al., 2012); ii) brownish crystals that cover the entire morphological range, which is induced by non-crystallographic branching; and iii) aggregates of the two aforementioned types (**fig. S6**).

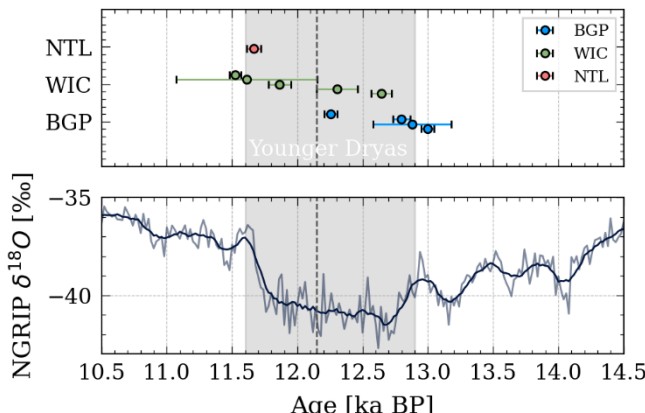

**Fig. 2: Formation ages of CCCs in comparison to other proxy records.** *Upper panel*: CCC formation ages from the studied cave sites. Error bars refer to the 2σ error of the respective isochron. Grey shaded area represents the defined limits of the Younger Dryas period. Caves are sorted by latitude (north is top) along the vertical axis and the position of individual data points is offset arbitrarily for better visibility. *Lower panel*: δ¹⁸O in Greenland precipitation from the NGRIP ice-core record (Rasmussen et al., 2014). The smoothed line represents a 100-year moving average. The dashed vertical line indicates the mid-Younger Dryas transition (12.15 ka BP).

Stable isotope analyses of the studied CCC deposits yielded values between -16.0 and -8.1 ‰ for $\delta^{18}O$ and -7.4 and -0.2 ‰ for $\delta^{13}C$ (**fig. S7**). These values follow a characteristic offset from common dripstone/flowstone speleothems in the area, which have significantly higher $\delta^{18}O$ values of approximately -5 to -3 ‰ and lower $\delta^{13}C$ values of approximately -11 to -8 ‰ (**fig. S7**; e.g., Gunn et al., 2020). This offset between common speleothems and CCCs is diagnostic for 'CCC$_{coarse}$' (sensu Luetscher et al., 2013), hence, we can be confident that our samples formed during the slow freezing of water in a stable climate (Žák et

al., 2018).

   Overall, 51 U/Th analyses were performed on 10 distinct CCC deposits from the three caves in the study area (**table S1**) and an isochron age for each CCC deposit was calculated. The formation ages of CCC deposits cover a timespan from 13.00 ± 0.05 to 11.52 ± 0.04 ka BP (**table 1**), which is broadly consistent with published definitions of the Younger Dryas boundaries (Cheng et al., 2020). Ages from Bagpit cave cover only the earlier part of the Younger Dryas (c. 13.00 to c. 12.26 ka BP),

whereas the analysed CCC deposits from the two caves in the north are more concentrated towards the latter half of the period (c. 12.64 to c. 11.52 ka BP) (**fig. 2**). Overall, however, there is no prominent clustering of ages.

**Ground Temperatures during the Younger Dryas**

We infer from temperature monitoring data at the cave sites that conductive heat flux is the dominant mode of energy transfer, and cave temperatures can be assumed to mirror changes in atmospheric temperature on decadal or longer timescales. The

distribution of CCC ages suggests that ground temperatures (in a depth range between 20 and 130 m) around the respective cave sites were within the CCC formation window throughout large parts of the Younger Dryas. Note that in this study, we

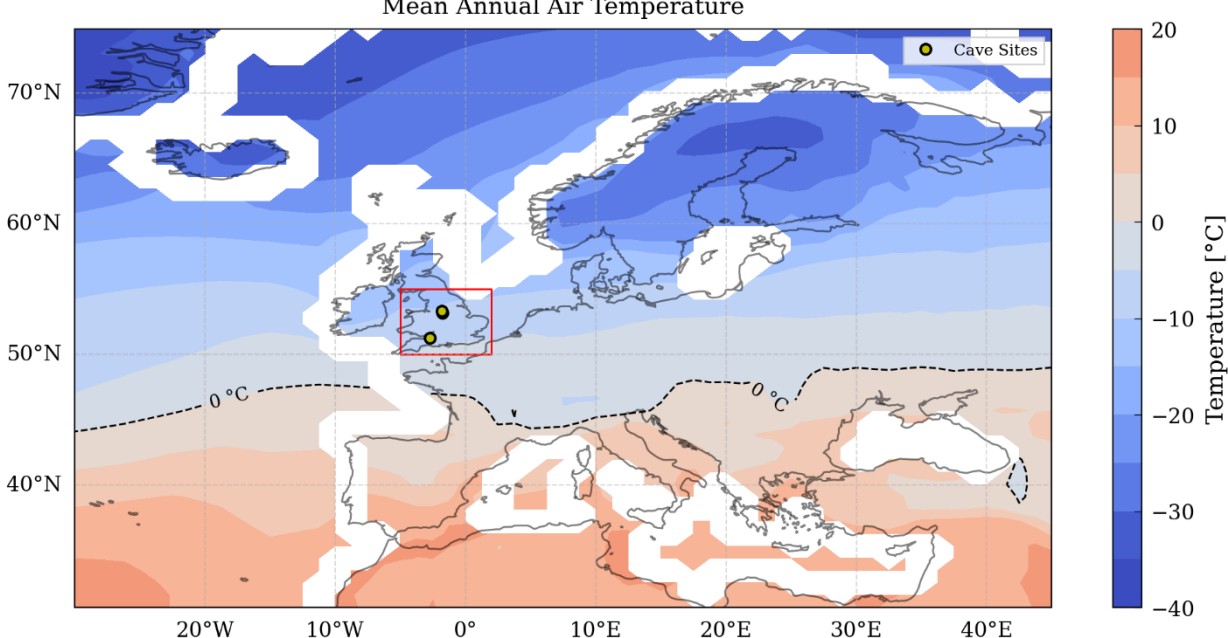

**Fig. 3: Mean annual air temperature over Europe according to a high-resolution model simulation of the Younger Dryas (Schenk et al., 2018). The positions of cave sites in this study are depicted as yellow dots and the red rectangle delineates the study area (-5 to 2 °E, 50 to 55 °N).**

conservatively opted to extend the CCC formation window to -2 – 0 °C as opposed to previous studies (i.e., -1 to 0 °C in Koltai et al., 2021; Spötl et al., 2021) in order to account for short-term, positive thermal anomalies that may arise from convective heat transfer by substantial amounts of infiltrating water.

The presence of permafrost in the study area during the Younger Dryas has already been demonstrated in previous studies. For example, geomorphological and sedimentological evidence from various sites across northern Europe suggests that continuous permafrost (MAAT < -8 °C, see **fig. 1**) existed north of 54 °N, with discontinuous permafrost (MAAT < -1 °C) reaching as far south as 50 °N (Isarin, 1997). Moreover, the Younger Dryas coincides with a hiatus of burial sites in the archaeological record of the British Isles which is possibly linked to changed funerary practices in response to perennially frozen ground (Blockley

and Gamble, 2016). These results from independent archives align well with the British CCC record.

Critically, CCC records are inherently discontinuous in time. A given CCC deposit can only provide evidence for the presence of permafrost at a certain point in time, whereas the absence of CCCs does not verify the absence of permafrost. Consequently, it is necessary to take extreme caution when interpreting gaps in the CCC record. Precise temporal constraints for periods of permafrost aggradation and/or degradation cannot be derived directly from CCC ages. It is possible that significant, short-term

spikes in temperature occurred during the gaps in the British CCC record (**fig. 2**), although there is no evidence in other (continuous) proxy records to suggest they did (e.g., Brauer et al., 2008; Lane et al., 2013; Cheng et al., 2020).

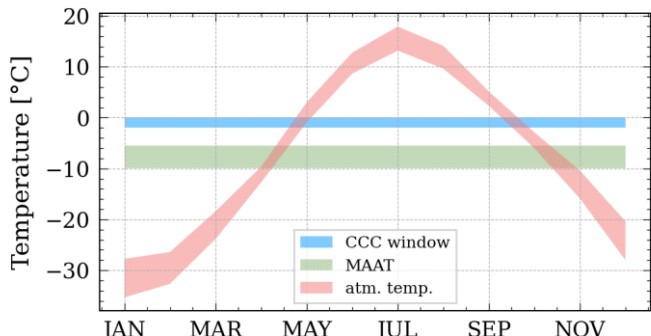

**Fig. 4: Simulated temperature seasonality for the Younger Dryas averaged over the study area (red rectangle in fig. 3). In the absence of insulating layers, ground temperatures below the depth of zero annual amplitude would equilibrate to the MAAT of -7.6 ±2.1 °C, which is well below the zone of CCC formation.**

Interestingly, a large number of climate records from the North Atlantic realm and elsewhere show a prominent switch from an extremely cold and dry climate to notably milder conditions that occurred mid-Younger Dryas at approximately 12.15 ka BP (e.g., Brauer et al., 2008; Bakke et al., 2009; Lane et al., 2013; Baldini et al., 2015; Cheng et al., 2020). The British CCC record also shows no data in the southern sector after 12.21 ka BP, though it is difficult to assess its significance, especially because the dataset (1 cave with 4 CCC deposits in this case) is small. However, our data do not exclude the possibility of a northward migration of the ground 0 °C isotherm at the time of the mid-Younger Dryas transition, nor do they support it. Overall, the British CCC record provides convincing evidence that ground temperatures at the studied cave sites were in the range of the CCC formation window between -2 and 0 °C throughout most of the Younger Dryas Stadial.

**Common Speleothems during the Younger Dryas**

By counterargument, the presence of permafrost as derived from the British CCC record would imply a cessation of growth of common speleothems. The published literature gives ambiguous evidence of a general Younger Dryas speleothem hiatus. Numerous studies report speleothem ages that at least overlap the Younger Dryas boundaries within error (e.g., Atkinson et al., 1978; Gascoyne et al., 1983; Atkinson et al., 1986; Baker et al., 1996). However, these studies mostly used alpha-spectrometry methods, which have large analytical uncertainty compared to modern mass-spectrometry-based methods (i.e., between 7 and 150 % of the absolute age at the 1σ level, which corresponds to an age uncertainty between ± 0.8 and ± 18 ka in absolute numbers for a 12.0 ka old sample). Due to this excessive age uncertainty, these data are not suitable to detect a potential speleothem hiatus during a millennial-scale climate event such as the Younger Dryas. While numerous Younger Dryas speleothem records have been published from continental Europe south of the study area, modern mass-spectrometry-based evidence for common speleothem growth on the British Isles has only been reported from two caves in southern Ireland: Crag Cave (Swabey, 1996; McDermott et al., 1999; Fankhauser et al., 2016) and Mitchelstown Cave (Swabey, 1996). These sites are located at 52 °N, roughly equivalent to the southernmost site (BGP) reported in this study, albeit in closer proximity





**Table 2: Younger Dryas temperature proxies.** Compilation of published estimations for Younger Dryas MAAT at the cave sites from paleo-glacier ELA (Rea et al., 2020), the mutual climatic range of Coleoptera remains (Atkinson et al., 1987) and periglacial features (Isarin, 1997). The temperatures were adjusted to the respective cave elevation with a lapse rate of -6.5 °C km⁻¹.

| Cave Site | Latitude | Longitude | Elevation | Younger Dryas MAAT estimation [°C] | | | Mean |
|---|---|---|---|---|---|---|---|
| | | | | Rea et al. | Atkinson et al. | Isarin 1997 | |
| | [°N] | [°E] | [m a.s.l.] | 2020 | 1987 | (central value) | [°C] |
| Water Icicle Close Cavern | 53.178 | -1.760 | 341 | -7.9 | -5.6 | -8.6 | **-7.4 ±1.3** |
| Nettle Pot | 53.335 | -1.813 | 467 | -9.0 | -6.4 | -9.4 | **-8.3 ±1.3** |
| Bagpit | 51.267 | -2.727 | 257 | -4.1 | -5.1 | -4.6 | **-4.6 ±0.4** |
| | | | | | | | **-6.7 ±0.7** |

to the Younger Dryas coastline (**fig. 1**) and at significantly lower elevation (approximately 100 m a.s.l). To date, there is no conclusive evidence of common speleothem deposition during the Younger Dryas either in the caves reported herein, nor in
any of the caves in their proximity. Given that the ground temperatures we inferred from CCCs are close to 0 °C, it would, however, not be surprising to see at least episodic common speleothem growth at other caves where conditions such as aspect, elevation or the cave ventilation regime were more favourable, like at the cited examples in Ireland. Overall, the available evidence from common speleothems in connection with our CCC ages is in good agreement with the distribution of continuous and discontinuous permafrost inferred from periglacial features (Isarin, 1997, see **fig. 1**).

**Temperature Offset between Permafrost and the Atmosphere**

Periglacial activity, including thermal contraction cracking, was widespread during the Younger Dryas in the British Isles (Ballantyne and Harris, 1994; Murton and Ballantyne, 2017). Based on observations from today's permafrost regions, ice wedges tend to crack by thermal contraction most frequently in areas with a MAAT of lower than about -6 and -8 °C in coarse-grained sediments (sand and gravel) and lower than about -4 °C in fine-grained sediments (silt and clay), although these
constitute only approximate upper estimates because local factors also influence cracking (Ballantyne, 2018b). At the time of cracking, cold (winter) air temperatures of usually between -25 to -40 °C, and ground-surface temperatures from -15 to -25 °C are considered necessary (French, 2017). The mean atmospheric temperatures which prevailed throughout large parts of the Younger Dryas have been constrained by numerous quantitative proxy records. Here, we are using the results of meta-analyses which in total compiled temperature estimates from over 300 individual sites in the British Isles and adjacent NW Europe
(locations shown in **fig. 1**) to derive a robust estimate of Younger Dryas MAAT:

A reconstruction based on the equilibrium line altitude (ELA) of paleo-glaciers on the British Isles found that the MAAT in the study area (50–55 °N) was between 0.3 and -7.9 °C (Rea et al., 2020). A record of MAAT based on the mutual climatic range of Coleoptera (i.e., beetles) shows comparable values of -3 ±2 °C for the Younger Dryas (Atkinson et al., 1987).



Furthermore, MAAT estimates derived from periglacial features attributed to the same time period provide a temperature range

of -1 to -8 °C (Isarin, 1997). After adjusting these MAAT estimates to the elevation of our cave sites by applying an environmental lapse rate of -6.5 °C km⁻¹, we determined that the resulting averages of MAAT values at the cave sites are -7.4 ±1.3 °C for WIC, -8.3 ±1.3 °C for NTL and -4.6 ±0.4 °C for BGP, giving a total average for all cave sites of -6.7 ±0.7 °C (**table 2**).

In this study, we compare CCC-derived mean annual ground temperature (MAGT) at depths of 20 to 130 m to atmospheric

surface temperatures from a recent modelling study using the Community Earth System Model (CESM 1.0.5). The model was run at a higher spatial resolution than other global circulation models for the Younger Dryas period (i.e., 0.9° × 1.25° or approximately 100 x 100 km) and the resulting surface temperatures are in good agreement with a wide range of available proxy records (Schenk et al., 2018). **Figure 3** shows the modelled Younger Dryas MAAT distribution over Europe according to CESM simulations. The study region (-5 to 2 °E, 50 to 55 °N, red rectangle in **fig. 3**) shows a MAAT value of -7.6 ±2.1 °C

(the uncertainty refers to the 1σ spatial variability of MAAT across the grid cells within the study area), which is in agreement with the proxy-derived MAAT for our study sites (**table 2**). Modelled monthly temperature averages cover a range from -31.6 ±3.7 °C in January to 15.5 ±2.4 °C in July, resulting in an hypercontinental climate with extreme temperature seasonality of 47.1 ±4.4 °C (**fig. 4**).

Such surface temperatures are certainly cold enough to facilitate permafrost formation in the study area during the Younger

Dryas. However, the modelled MAAT (-7.6 ±2.1 °C) as well as the averaged MAAT reconstructed from proxies (-6.7±0.7 °C, **table 2**) is significantly colder than the CCC-derived MAGT (-1 ± 1 °C), with an offset of ΔT = 6.6 ±2.3°C (CESM) or ΔT = 5.7 ±1.7°C (proxy reconstructions), respectively. The relationship between MAAT and MAGT in permafrost is non-linear and observed offsets are the result of multiple factors, including aspect, snow cover, vegetation, seasonality, soil moisture and the porosity of the substrate (Murton, 2021). In principle, two types of offsets can be distinguished (Smith and Riseborough, 2002,

see **fig. 5**):

1. The *nival offset* (ΔT$_S$) includes the insulating effects of snow cover. Since snow is a poor conductor of heat, sufficiently thick snow cover limits heat loss from the ground during winter and causes a positive offset of the MAGT from MAAT. The nival offset only has a warming effect while air temperatures are below 0 °C (i.e., when snow is not melting).

2. The *thermal offset* (or active-layer offset, ΔT$_T$) results from higher rates of heat conduction through ice as opposed to liquid water or air. In winter, when moisture in the soil and epikarst layer has turned to ice, heat conduction is more efficient (i.e., approximately four times faster) than in summer, thus creating a net cooling effect that counteracts the nival offset.



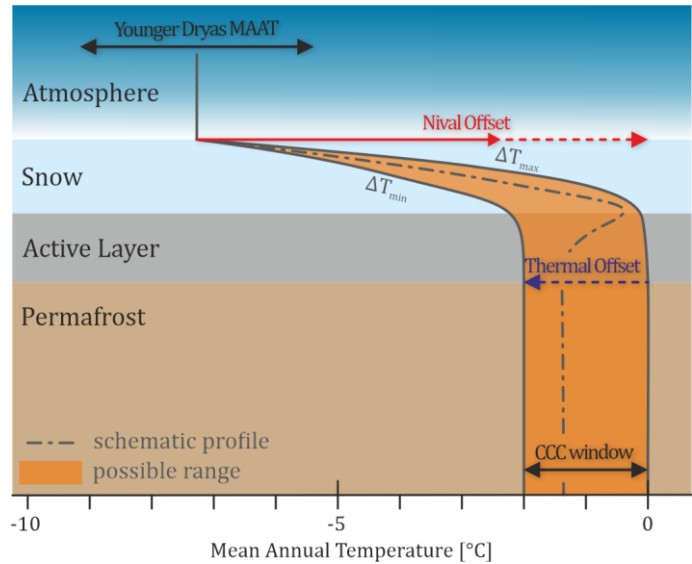

**Fig. 5: Conceptual model of permafrost temperature in relation to MAAT and snow thickness, modified after Smith and Riseborough (Smith and Riseborough, 2002). Depicted values for the Younger Dryas MAAT and the CCC window for permafrost temperature are as mentioned in the main text. Solid grey lines correspond to the inferred minimum and maximum total offset values $\Delta T_{min}$ and $\Delta T_{max}$, assuming that the thermal offset is 0. Since this assumption is unlikely, any profile (schematic example shown as dash-dotted grey line) within the limits of these endmembers is theoretically possible.**

Our CCC-derived MAGT and modelled MAAT provide quantitative constraints on the net effect of these temperature offsets.

With the warming contribution of the nival offset limited to a maximum temperature of 0 °C, the maximum $\Delta T$ in the absence of a corresponding thermal offset is $\Delta T_{max} = 7.6 \pm 2.1$ °C. Conversely, MAGT could not have been cooled below the CCC formation window (-1 ± 1 °C), resulting in a minimum offset of $\Delta T_{min} = 5.6 \pm 2.1$ °C. These values are considered endmembers and delineate a range of possible temperature profiles that arise from different combinations of nival and thermal offsets but result in $\Delta T$ values within the specified range (**fig. 5**).

The relatively narrow range of possible temperature profiles implies that the warming contribution of the nival offset at the study sites was much larger than the corresponding cooling attributed to the thermal offset within the active layer. A minor cooling effect via the thermal offset is consistent with situations at the study sites where either soil moisture was low because of good drainage through the underlying karst system, or simply because the soil layer was not thick enough to have a significant effect. Of course, the paleo-active-layer thickness at the three cave sites is unknown, though by analogy with

relatively warm discontinuous permafrost today in the mountains of northern Norway, where the MAGT at 10 m depth is near 0 °C and the bedrock is relatively dry (Farbrot et al., 2013) and so the thermal offset is probably limited, it may have ranged in the order of 10 m. Either way, the observed range of $\Delta T$ points towards significant seasonal snow cover as the main cause of temperature offset in these limestone terrains of the British Isles during the Younger Dryas.





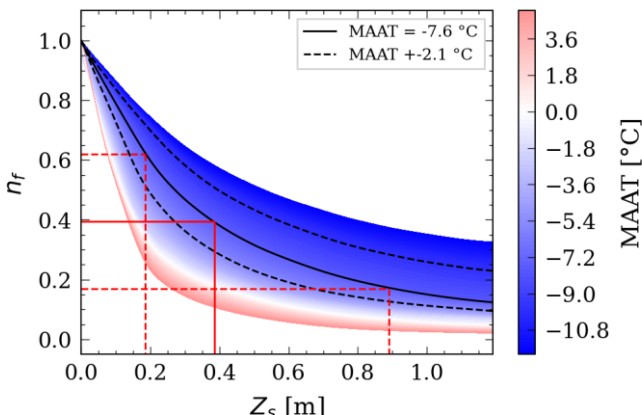

**Fig. 6: Snow-thickness reconstruction according to the analytical approach. Depicted is the relationship between snow thickness ($Z_s$), snow cover n-factor ($n_f$) and MAAT according to Riseborough and Smith (Riseborough and Smith, 1998). The solid black line represents the modelled Younger Dryas MAAT at the studied cave sites with uncertainty boundaries shown as dashed black lines. Red solid and dashed lines represent the $n_f$ value calculated from the CCC-derived $\Delta T$ and its respective uncertainty. A mean annual snow thickness $Z_s$ between 0.2 to 0.9 m can be reconstructed, with the highest probability at approximately 0.4 m.**

**Younger Dryas Snow Cover and Modern Analogues**

By combining CCC-derived MAGT and modelled temperatures, it is possible to constrain the temperature offset between permafrost and the atmosphere at the study sites during the Younger Dryas and to identify snow cover as the crucial factor. We apply two principal lines of argument to quantify the amount of snow cover that best explains the observed $\Delta T$.

   1. **Analytical approach**

      Smith and Riseborough (2002) described the relationship between MAAT and MAGT with a set of equations

employing air temperature indices and n-factors to assess the contributions of the nival and thermal offsets, also known as the *TTOP (temperature at the top of permafrost) model*. Re-arranging these equations, the n-factor corresponding to the snow cover effect $n_f$ can be expressed as

$$n_f = \frac{\Delta T_S \cdot 365}{I_f} \tag{1}$$

where $I_f$ is the annual air freezing index (i.e., degree days <0 °C), which can be derived from the MAAT via an empirical relationship

$$I_f = -241.6 \cdot MAAT + 2142 \tag{2}$$

Using the value we derived for $\Delta T = 6.6 \pm 2.4$ °C as input for $\Delta T_S$, eq. 1 gives $n_f = 0.39 \pm 0.22$ as the corresponding range of n-factors for the nival offset at the study sites during the Younger Dryas. This n-factor relates to the mean





annual snow thickness $Z_s$ according to another relationship provided by Riseborough and Smith (1998) that is depicted in **figure 6**. With the values for $n_f$ and MAAT that derive from modelling and CCCs, a mean snow thickness for the study sites during the Younger Dryas between 0.2 and 0.9 m can be reconstructed, with a nominal value of approximately 0.4 m, representing a snow water equivalent (SWE) of 148 ±125 mm assuming a bulk snow density of 0.27 ±0.15 g cm$^{-3}$ (Sturm et al., 2010).

2. **Statistical approach using modern analogues**

We use 30-year averages for air temperature 2 m above the ground surface, MAGT, snow cover days (SCD) and snow height from reanalysis data (ERA5 monthly from 1990 to 2020; Muñoz Sabater, 2019) to quantify the relationship between MAAT, MAGT and snow cover in areas of the Northern Hemisphere where permafrost currently exists (**fig. 7**, see methods section for a detailed explanation). From the quantitative constraints we derived from CCCs and CESM simulations (i.e., MAGT of -1 ±1 °C and MAAT of -7.6 ±2.1 °C; red rectangle in **fig. 7a**), it is possible to identify those grid cells in our reanalysis dataset that best resemble the British Isles during the Younger Dryas with respect to surface and ground temperatures.

The geographical locations of these *modern analogues* can be plotted on a map as shown in **figure 7c**. They predominantly cluster at low-elevation, coastal areas bordering the Arctic Ocean (e.g., Yamal Peninsula, NE Siberia, NW Alaska, and parts of the Arctic Archipelago) with some minor clusters at inner-continental mountainous areas (e.g., Mackenzie Mts. and Altai Mts.). On average, the mean annual snow cover at these locations lasts for 233 ±54 SCD per year, whereas the SCD distribution of this subset is very similar to the general distribution of the whole reanalysis dataset (**fig. 7b**). The corresponding mean snow thickness as recorded in the reanalysis data from 1990 to 2020 is 0.5 ±0.2 m or 135 ±92 mm of SWE with the same range of bulk snow densities as used in the analytical approach.

The two methods used for assessing the potential range of snow thickness based on thermal parameters derived from the British CCC record are independent of each other. The results are in excellent agreement, albeit with substantial uncertainty that arises from Younger Dryas temperature estimates and a lack of constraints on other critical parameters such as snow density. The uncertainty does, however, fall within the range of inherent year-to-year snow thickness variability.

It needs to be explicitly stated at this point that the snow thickness estimates are not equivalent to the total winter precipitation amount and the two should not be compared directly without applying adequate transfer functions. The validity of our snow thickness estimates may be assessed qualitatively against paleo-precipitation estimates for the Younger Dryas. Most quantitative estimates of Younger Dryas paleo-precipitation in the British Isles come from reconstructions of glacier ELAs. In Scotland, north of our study area, these reconstructions infer a steep west-to-east decline in precipitation (Ballantyne and Harris, 1994; Palmer et al., 2010). Annual precipitation from the Monadhliath ice field adjusted to sea-level equivalent gives an annual precipitation of 1211 ±480 mm, 692 ±360 mm of which fell as snow. Winter precipitation values from other glaciers

cited in this study are between 379 ±315 and 2066 ±323 mm. Throughout the winter season, the snowpack is subject to

compaction and erosion processes that generally reduce the overall thickness. Consequently, the mean SWE is always lower than the corresponding winter precipitation amount over the course of the accumulation season. The Younger Dryas SWE estimates we inferred are consistently lower than glacier-derived winter precipitation values from Scotland and can be regarded as valid estimates according to this logic.

**Climatic Context**

Winter sea-ice inhibits heat and moisture exchange between the ocean and atmosphere during the cold season, thus contributing to extreme winter cold, while summer temperatures and precipitation might remain largely unaffected (Singarayer et al., 2006). Numerous studies have proposed a model where the North Atlantic experienced a surge in seasonal sea-ice during the Younger Dryas, which in turn caused extreme temperature seasonality over downwind land masses (e.g., Atkinson et al., 1987; Renssen and Isarin, 1997; Isarin and Renssen, 1999; Denton et al., 2005; Schenk et al., 2018). While the causal relationships are still

under debate, most authors agree that the fundamental mechanism behind this surge is a weakening of the AMOC and southward migration of the sea-ice edge in winter down to ~45°N (Schenk et al., 2018).

At this point, the geographical location of the modern analogues we identified with our statistical approach warrants further discussion. The (coastal) modern analogues we inferred from the British CCC record border parts of the Arctic Ocean that experience seasonal sea-ice cover in the modern-day climate setting (**fig. 7c**). More precisely, they are situated in the vicinity

of the 90% contour line of winter (DJF) sea-ice concentration and are far away from perennial sea-ice.

The extreme seasonality of 47.1 ±4.4 °C that is implied in the CESM simulations we used in our study is in agreement with the aforementioned model of a Younger Dryas sea-ice surge, whereas we found that significant amounts of seasonal snow cover are necessary to reconcile the modelled air temperature with MAGT we empirically derived from CCC deposits. The fact that the modern analogues we identified via the same approach also experience seasonal sea-ice cover stresses that winter

sea-ice in the North Atlantic reached the study area at 50 to 55 °N during the Younger Dryas consistent with a simulated maximum extent down to ~45°N in CESM (Schenk et al., 2018).

Empirical evidence for extensive North Atlantic sea-ice during the Younger Dryas is provided by marine sediment cores (Cabedo-Sanz et al., 2013; Müller and Stein, 2014) and was also suggested on the basis of multiple terrestrial proxy records (e.g., Brauer et al., 2008; Lane et al., 2013; Baldini et al., 2015). A connection between sea-ice and long-term permafrost

stability has been proposed for Siberia in previous studies (Vandenberghe et al., 2012; Vaks et al., 2020) and is consistent with the British CCC record. Overall, our study supports a model of the Younger Dryas as a period of extreme seasonality dominated by severe and long-lasting winters in response to increased winter sea-ice extent and amends the existing body of evidence with empirical constraints on ground temperatures and snow-cover thickness and duration.





**Fig. 7: Visualization of the statistical approach using modern analogues. (A) Relationship between the modern-day temperature offset between MAAT at 2 m above ground surface, MAGT and snow-cover days (SCD) from reanalysis data (ERA5 monthly from 1990 to 2020). Individual data points represent 30-year averages of a gridded subset that covers areas of the Northern Hemisphere where the permafrost fraction is larger than 10 %. The red rectangle delineates the range of values constrained by CCC-inferred MAGT and CESM simulations for the Younger Dryas in the study area. (B) Normalised kernel density estimates of SCD of the same dataset depicted in (A) (i.e., all data) and the subset constrained by the CCC- and model-inferred MAAT and MAGT (i.e., YD analogues). (C) Map of modern permafrost distribution (*Obu et al., 2021*, data for year 2010). Blue colors show the 30-year mean sea-ice concentration between 1987 and 2017 (Walsh et al., 2017). The black and yellow lines delineate the mean position of the 90 % concentration contour line during winter (DJF) and summer (JJA) respectively of the same 30-year observation period. Black dots show the location of grid cells that fall within the CCC-derived temperature window shown in (A).**



## Code/Data availability

All raw data necessary to replicate the results presented in this study are provided in the main text, the supplementary material or are available via the cited sources. The code used for the various calculations is available upon request from the corresponding author.

## Competing interests

The authors declare no competing interests.

## Author contribution

PT and GEM designed the study, conducted field work and performed laboratory analyses. PT and AB conducted data post-processing analyses. FS designed and ran the CESM 1.0.5 climate simulations and together with JBM was involved in data analyses and data-model integration. PT wrote the initial draft and all co-authors were involved with supervision and editing of the manuscript.

## Acknowledgements

The authors thank Xianglei Li, Dylan Parmenter and Peter Schroedl for assistance with U/Th analyses as well as Lena Friedrich for additional stable isotope analyses. The help of Alison Moody, John Gunn, Alan Brentnall, Rob Eavis, Andy Freem and Robbie Shone with cave access, guiding, additional fieldwork, surveying and discovering CCC sites is greatly appreciated. Sincere gratitude is extended towards Natural England and the Trustees of the Devonshire Maintenance Fund for permission to conduct fieldwork and collect samples. We also thank Dr. Sina Longman for providing us with access to her extensive database of speleothem ages from the British Isles, which pleasantly streamlined the literature review process.

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
