# Peer review of "Reconstructing Younger Dryas Ground Temperature and Snow Thickness from Cave Deposits"

_Climate of the Past, 2023_

## Author Response (AR1)

**Response to RC1**

*We thank reviewer 1 for the diligent, concise, and constructive assessment of our manuscript. We very much appreciate the effort! Please find our response below.*

My first "major" comment concerns the uncertainty propagation of the temperature (offset) estimates. It is unclear to me how propagated uncertainties are calculated (L241ff). While the temperature offset uncertainty for the proxy reconstructions appears to be calculated by the sum of the errors of MAGT and MAAT, this seems to be different for the offset for the CESM. Likewise, in L280 a different uncertainty for the temperature offset is given (most likely a typo), but please clarify.

*We've used a python package called "uncertainties" to do the error handling based on linear propagation theory (https://pythonhosted.org/uncertainties/). We'll add a sentence to the respective method section to clarify this. We also re-calculated all error propagations and found rounding errors in table2 (mean = -6.7 ±0.6, not ±0.7) which has consequences for line 242 (±1.2 instead of ±1.7) – thanks for drawing our attention to this!*

Secondly, I would also ask the authors to provide more context on the involved models and methods to estimate Younger Dryas snow cover (L270ff). The model used in part (1) ("Analytical approach") seems to be kind of a black box, and it would be good to include more explanations of the components and assumptions of the model. In part (2) I was wondering if the grid cells used as modern analogues were also checked if they are "analogues" also in terms of seasonality? The seasonal T differences appear to have been extreme according to the model, so this could also affect the results.

*(1) was also mentioned by the second reviewer and we'll add a brief summary of the TTOP model at the beginning of this section. For point (2): Yes, these grid cells also show extreme seasonality – we'll add in a bit about the seasonality/continentality of the modern analogues in comparison to our study sites during the YD.*

Lastly, I miss a more summarizing, concluding statement, what the results actually mean? For example, how "new" or "(un-) expected" are the results? The current version of the manuscript leaves the reader a bit lost on how to assess the numbers of snow days per year or the derived snow thickness etc...

*Both reviewers seem to agree on this part as well – we consider adding a conclusions section at the end.*

In addition, I have some minor comments along the text:

L17/18: I would provide a statement/context in the abstract, what these results actually mean.

*We'll add a sentence to summarize the conclusions.*

L23 Maybe state a number of what is meant by "extremely cold" of "most severe cooling"

*We intentionally did not put hard numbers in this early part of the manuscript because the regional estimate for the Younger Dryas cooling is a matter of ongoing (and as we've come to realise, quite lively) debate. Since we dedicate a large portion of the discussion to come up with a composite "best-guess" temperature estimate, we feel that it would be overly simple to state a number at this point. What we can offer is to cite a range of high and low estimates from the literature.*

L38ff Sometimes it is unclear, if the statements refer only to Britain, Northern Europe in general, or other areas. This relates a bit to the whole introduction, that it could be more clearly outlined how well YD climate/permafrost is investigated in Britain, in N Europe, and why it would be important/necessary to find out more about it…

*We'll screen the introduction for those specific cases and clarify where appropriate.*

Fig 1: Colors of the cave symbols are hard to see

L93ff Please provide also absolute min/max temperatures for the cave

*We'll fix that.*

L100 does this impact ventilation / temperature exchange?

*We'll add a sentence along the lines of "While excavations may have altered air circulation patterns, monitoring shows that the cave temperature at the respective sites are unaffected."*

L163ff Would it be possible to elaborate this in a bit more detail? In the site description, it is noted, that convective heat flux is not dominant for Nettle pot?

*There seems to be a misunderstanding. In line 88-89 we state that "[...] a thermal anomaly by convective heat fluxes, [...] is unlikely." In other words, conductive heat flux is likely dominant. We don't see a contradiction with the statement in line 163 that "[...] conductive heat flux is the dominant mode of energy transfer, and cave temperatures can be assumed to mirror changes in atmospheric temperature [...]". We modified the sentence to make this more clear.*

L241 How is the error estimated? Shouldn't it be 3.1°C? (2.1 + 1°C)

*See comment above. According to linear propagation theory, $A = B+C$, the corresponding uncertainties propagate as $\delta A = sqrt(\delta B^2 + \delta C^2)$, where $\delta$ denotes the respective uncertainty. So in the given case of subtracting the CCC derived ground Temperature and the modelled MAAT this means $\delta(T_{CCC} - MAAT_{modelled}) = sqrt(1^2 + 2.1^2) = \pm 2.3$*

L256 see previous comment (uncertainty?)

*See above.*

L257 also here related to previous comment (uncertainty?)

*See above.*

L274ff only one sentence to explain the model that produces the main result is very few information…

*See above – we'll add a short summary of the TTOP model here.*

L280 now its +- 2.4°C?

*Good catch – that appears to be a typo. The result for nf is correct.*

L292 where these grid cells also checked if they compare well in terms of seasonality, i.e. min/max temperatures?

*See above*

**Response to RC2**

*We also extend our gratitude to reviewer 2 for providing a helpful and honest assessment of the manuscript. The time and effort is very much appreciated.*

The authors have completed a really nice study on cryogenic cave carbonates from three sites in the British Isles that constrain mean annual ground temperatures (MAGT) during the Younger Dryas interval. The CCC, compared to other estimates of mean annual air temperatures during the YD suggest that the ground must have been significantly warmer than the air, and a model suggests that a snow cover thickness of around 0.4 m over a long winter season could allow ground temperatures in the CCC thermal window. These data add evidence for YD conditions in the British Isles and North Atlantic region during a critical time interval in the last deglaciation. The authors provide some modern analog locations similar to what they estimated for the YD.

Overall, the results are solid and support the authors interpretation. I cannot identify any problems with the paper. The writing was clear (mostly, but see below), and the reasoning solid.

Perhaps the only substantive comment I have on the manuscript is that the paper ends abruptly at line 343. It seems to be missing a conclusion, or at least a paragraph to summarize the main findings and implications of the study. The authors might consider some broader questions related to their data in a summary paragraph of section.

*This was also pointed out by reviewer 1 – we will add a short conclusions section at the end.*

Figure 1. Indicate that the isotherms are estimate YD isotherms. I got confused when I saw the figure because I assumed they were modern initially but I had to read carefully to find out not.

*We can add this information to the legend box.*

In the Modern Climate Data section, I feel there needs to be a better description of the permafrost data: what it is, why it was used, what time it represents. Is it only for the Britain? The study sites? Eurasia?

*We'll add 1-2 sentences summarizing the paper that accompanies this data set.*

Figure 2. Records, plural? I only see one other paleoclimate record. Add the Cheng et al 2020 citation to caption.

*Good catch – that's an artefact from earlier versions. We'll fix that.*

Fig. 6 caption, I think it should read ±2.1 not +-2.1

*Another good catch and we'll fix that too.*

To a reader not versed in the details of the TTOP method, it would be helpful to walk the reader through it using plain English.

*This was also pointed out by reviewer 1 and we agree there should be a brief summary of the TTOP model somewhere around line 250.*

Can the authors say anything about MAGT during the preceding Bolling period? It is interesting that the CCCs correspond so well to the YD, but what would have been the MAGT depression during the YD relative to the Bolling?

*Certainly, a good question but – as we tried to emphasise from line 176 onwards – interpreting a perceived hiatus of CCCs can be a red herring. Out data can't add anything substantial to this question other than speculations. We'd rather stick to the part of the deglacial where we have good data – which sadly is only during the Younger Dryas.*